# Intra-Seizure Pattern Recognition for Personalized Treatment

Saeed Hashemi, Genchang Peng, Mehrdad Nourani[*]
*Department of Electrical and Computer Engineering*
*The University of Texas at Dallas*
Richardson, TX 75080
{Saeed.Hashemi, gxp170004, nourani}@utdallas.edu

Omar Nofal, Jay Harvey
*Department of Neurology*
*The University of Texas Southwestern Medical Center*
Dallas, TX 75390
{Omar.Nofal, Jay.Harvey}@UTSouthwestern.edu

*Abstract*—Stereo-electroencephalography (SEEG) is a technique to monitor and evaluate the spatial and temporal properties of ictal EEG changes in patients with epilepsy during pre-surgery evaluation. When patients have different types of seizures, the intra-seizure patterns across different ictal episode provide extra and meaningful information for personalized treatment planning. This work introduces a patient-specific framework to capture the intra-seizure patterns in a seizure-specific way. After defining a Pre-seizure Plus Seizure (PPS) window as period of interest, SHapley Additive exPlanations (SHAP) is applied to quantify the contributing score of each SEEG channel (i.e. spatial correlation) based on XGBoost classifier. These SHAP scores are segmented and compared via Soft-Dynamic Time Warping (Soft-DTW) to characterize their dissimilarities (i.e. temporal pattern). Then, k-medoids clustering is exploited to divide seizure episodes into groups (episodes) based on Soft-DTW variations, and three clinically meaningful stages, namely trigger, transient, and steady stages, are consistently identified. Validated on SEEG data from eight patients, our results demonstrate high classification performance, reliable epileptogenic zone localization, and robust intra-seizure stage segmentation which are consistent with clinicians annotations.

*Index Terms*—Epileptogenic zone localization, Intra-seizure pattern, Seizure stage discovery, spatial-temporal representation, Stereo-electroencephalograpy.

## I. INTRODUCTION

### A. Motivation & Prior Works

Epilepsy is a neurological disease that affects millions of people worldwide [1]. Approximately one-third of patients are resistant to at least two anti-seizure medications, and are usually suggested to different epilepsy surgeries, laser ablation or neurostimulation [1]. To find the most effective treatment plan, a comprehensive pre-surgery evaluation must be performed using different techniques, including electroencephalography (EEG), neuroimaging, and sometimes invasive tools. Stereo-electroencephalography (SEEG) is a type of intracranial EEG (iEEG) that implants depth electrodes into the human brain to monitor and collect cerebral activities [2]. When traditional scalp EEG does not provide detailed information on surgical regions, SEEG is then applied to investigate deep brain activities for surgical planning with high resolution. SEEG has now been widely adopted in more than 300 institutions across the US for pre-surgery evaluation [2].

Accurate SEEG evaluation enables clinicians to localize and characterize the *epileptogenic zone (EZ)*, the hypothetical brain region responsible for ictal onset, propagating network

and symptomatic behaviors [3], in order to guide surgical resections or targeted neurostimulation [4]. The EZ exhibits both spatial and temporal characteristics: spatially, it includes *seizure onset zone (SOZ)*, the cortical areas where ictal EEG changes first appear; temporally, it reflects the dynamic evolution of different epileptic activities with clinical symptoms. Due to the high dimensionality of SEEG recordings, which involve 10 to 20 depth electrodes and up to 200 signal channels (contact points), data-driven SEEG analysis is applied to provide accurate and efficient insights beyond clinicians' visual inspections [2]. In the spatial domain, researchers have applied statistical analysis [4] and machine learning classification [5] to localize the seizure-contributing channels. In the temporal domain, researchers have quantified the signal patterns of different seizure stages (phases) from initialization, propagation to termination, utilizing dynamic seizure network modeling [6], [7] or deep neural networks [8], [9]. However, epilepsy patients may have different seizure types, and most of prior works treat seizure types equivalently during data analysis, ignoring the intra-seizure patterns, i.e., the varieties of ictal features from different seizure events in each individual patient.

Understanding intra-seizure patterns may assist in a more comprehensive and personalized plan. During pre-surgery evaluation, neurologists focus on the first 10 to 20 seconds of seizure-onset patterns [10] seen from the SEEG, neglecting late development of epileptic ictus involving symptomatogenic regions. Identifying the intra-seizure pattern could provide valuable information about those additional seizure-related regions beyond SOZ. EEG patterns may correlate with different clinical symptoms, e.g. focal seizures with preserved consciousness and impaired consciousness. In such cases, the information could contribute to the optimal EZ area and therefore surgical resection. Additionally, when responsive neurosimulation (RNS) is considered, stimulation parameters (e.g., pulse width and frequency) may be better defined in a personalized way. SEEG analysis that can correlate both spatial and temporal seizure patterns with a greater precision and clinical relevance continues to be the standard for pre-surgery evaluation. Researchers in [11] quantified the directional connectivities from frequency-domain seizure network and correlated with RNS outcomes. Authors in [12] adopted dynamic mode decomposition to extract the spatiotemporal dynamics of SEEG network for identifying EZ regions. In [13], a dynamic step effective network was employed to obtain the

---

[*] Corresponding author.

propagation pathways for surgical evaluation. However, these work required network modeling from SEEG data where parameter estimation (e.g., frequency band) could be challenging. Besides, they focused on the overall dynamics of entire seizure events, rather than intra-seizure stage discovery.

### B. Main Contribution

In this work, we introduce a patient-specific framework for SEEG-based seizure analysis that captures the spatial-temporal dynamics of individual seizure episodes. We train an XGBoost classifier to compute per-seizure SHapley Additive exPlanations (SHAP) and to quantify the contribution of each contact point (channel) during the Pre-seizure Plus Seizure (PPS) window. SHAP profiles act as temporal biomarkers, capturing how contributions of EZ evolve over time. To identify distinct seizure stages, SHAP temporal profiles are segmented and compared using Soft-DTW, a time-series-aware similarity measure. The resulting distance matrices are clustered using the k-medoids algorithm, enabling unsupervised discovery of consistent seizure stages without requiring manual annotations. We apply our analysis to each individual seizure, and consistently identify clinically relevant stages, i.e. mainly these three: i) an initial *trigger* stage, ii) an intermediate *transient* stage, and iii) a final *steady* stage. Findings offer a more detailed representation of seizure evolution and support future efforts to integrate stage-aware markers into personalized treatment, such as in pre-surgical evaluation or neurostimulation adjustment.

The remainder of this paper is organized as follows: Section II details the proposed methodology, including channel ranking, SHAP profiling, and clustering. Section III presents empirical findings and examines their alignment with clinical observations. Finally, Section IV summarizes the key contributions and outlines future directions.

## II. METHODOLOGY

This study, approved by the IRB at a Level-4 U.S. epilepsy center, includes SEEG from eight patients (3 male, 5 female, ages 18–52) with drug-resistant focal epilepsy, recorded at The University of Texas Southwestern Medical Center (UTSW). They overall produced 38 seizures recorded at 1000 Hz using a clinical EEG system. The overall methodology is summarized in Fig. 1 and explained next.

### A. Data Preprocessing

SEEG data were acquired for each patient based on clinical electrode implantation strategies (e.g., Fig. 2). For each seizure, a PPS window was extracted, encompassing both the seizure period and a pre-onset buffer to capture early seizure transitions. An illustration of the PPS segmentation approach is shown in Fig. 3, highlighting the seizure onset along with the extended pre-seizure region included in the analysis.

The analysis begins with channels selected by the clinical team based on their assessment of seizure onset, typically corresponding to the SOZ. To obtain a more comprehensive view, we adopt an electrode extension approach, as described in [14],

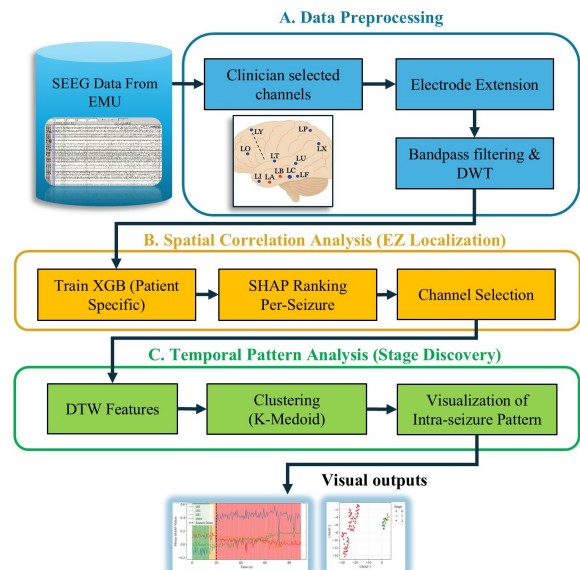

Fig. 1. Spatiotemporal analysis process with visual output of intra-seizure patterns.

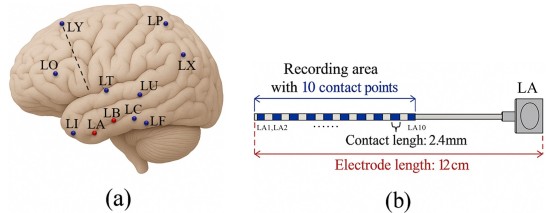

Fig. 2. SEEG example: (a) Implantation map with clinician-selected LA/LB in red; (b) ten contact points (LA1–LA10) on LA.

by including all contact points along the electrodes identified by clinicians. For instance, if LA1 is selected by clinicians, all channels LA1 through LA10 are analyzed to capture broader regional activity, as illustrated in Fig. 2(b). By expanding spatial coverage and incorporating temporal patterns, this method also facilitates assessment of the symptomatogenic region, providing a more complete characterization of epileptogenic involvement beyond the clinical-marked SOZ.

Signals were bandpass-filtered (1–60 Hz, 4th-order Butterworth) and decomposed using a Daubechies-4 DWT with five levels [15]. From each coefficient set (one approximation, five details), mean and standard deviation were extracted, yielding 12 features per channel per window. This representation effectively captures the dynamics of nonstationary SEEG signals.

### B. Spatial Correlation Analysis (EZ Localization)

An XGBoost classifier is trained per patient using all PPS and non-seizure windows, with an 80% - 20% (training-test) stratified split across seizures to ensure balance and prevent leakage. SHAP values [16] are computed per seizure to quantify each channel's contribution, forming the basis for selecting seizure-relevant channels and defining temporal stages. Classifier performance confirms its reliability for generating SHAP values. XGBoost, a gradient-boosted decision tree algorithm, is well-suited for structured biomedical data due to its scalability and high performance [17].

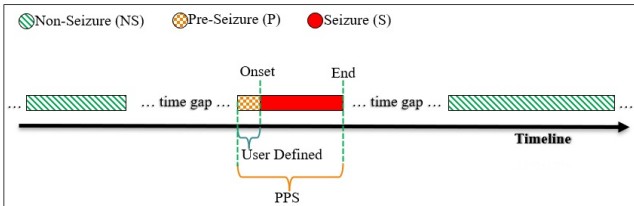

Fig. 3. Illustration of PPS segmentation, combining a user defined pre-onset buffer with the ictal period to capture early seizure dynamics.

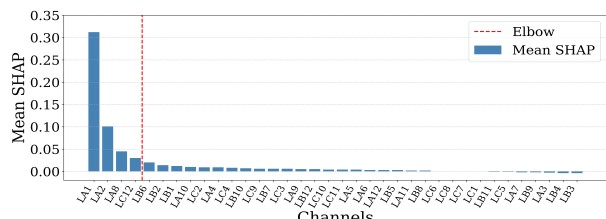

Fig. 4. SHAP ranking of LA, LB, and LC contacts (Patient 1000, Seizure 4). Red line = elbow cutoff; LA1–2 clinician-selected, LA8 and LC12 model-identified.

Given a channel $c \in N$, where $N$ is the set of input channels, the SHAP value at time $t$, denoted $\phi_{c,t}$, is computed as:

$$\phi_{c,t} = \sum_{S \subseteq N \setminus \{c\}} \frac{|S|!(|N| - |S| - 1)!}{|N|!} \left[ f_t(S \cup \{c\}) - f_t(S) \right], \quad (1)$$

where $S$ is any subset of channels excluding $c$, and $f_t$ represents the model's raw output at time $t$ (e.g., the unnormalized margin score before classification). This formulation ensures both local accuracy and consistent channel attribution over time [16].

To organize these values across all channels and time points, we define the SHAP matrix $\Phi \in \mathbb{R}^{C \times T}$, where $C$ is the number of selected channels and $T$ is the number of time segments within the PPS window:

$$\Phi = \begin{bmatrix} \phi_{1,1} & \phi_{1,2} & \cdots & \phi_{1,T} \\ \phi_{2,1} & \phi_{2,2} & \cdots & \phi_{2,T} \\ \vdots & \vdots & \ddots & \vdots \\ \phi_{C,1} & \phi_{C,2} & \cdots & \phi_{C,T} \end{bmatrix}$$

This matrix offers a compact spatial-temporal representation of SHAP dynamics and serves as the input for downstream segmentation and stage clustering within our framework.

In this work, we use 2-second window with 50% overlapping (1-second) to divide the SEEG signals into $T$ segments across the PPS. The mean SHAP value is then calculated for each contact point over $T$ segments, quantifying its overall contribution during the seizure. The elbow method [18] is applied to ranked SHAP scores to automatically determine the threshold, independent of clinician input. The SHAP ranking typically exhibits a sharp decline among the top-ranked channels, followed by a transition to smaller incremental changes. This inflection point defines a natural threshold and supports stable, data-driven channel selection. Fig. 4 shows the selection of 4 channels (LA1, LA2, LA8, LC12) from 36, with the elbow point at $C = 4$.

### C. Temporal Pattern Analysis (Stage Discovery)

To capture the evolution of seizure stages for each seizure individually, we analyze the temporal progression of SHAP values across the PPS window, where the three stages (trigger, transient, steady) are derived from clustering these temporal profiles rather than fixed amplitude or duration thresholds. These values are organized in the SHAP matrix $\Phi \in \mathbb{R}^{C \times T}$, where each row corresponds to a contact point and each column to a time segment. Since SHAP temporal profiles may differ in complexity across seizures, we first evaluate multiple window sizes (1–5 seconds) by pre-clustering with $k = 3$ for each window size. We then select the length that yields the highest silhouette score [19], reflecting intra-cluster compactness relative to inter-cluster separation. This selection ensures that the segmentation is adapted to the underlying seizure dynamics prior to clustering.

To compare multivariate temporal segments, we employ the Soft-DTW algorithm [20], a differentiable and robust variant of Dynamic Time Warping (DTW) that enables smooth alignment and real-valued cost aggregation over multidimensional sequences.

Let $X, Y \in \mathbb{R}^{C \times w}$ denote two temporal segments extracted from the SHAP matrix $\Phi$, where $C$ is the number of selected channels and $w$ is the chosen window length (in time steps). The Soft-DTW distance between $X$ and $Y$ is defined as:

$$\text{soft-DTW}(X, Y) = -\gamma \log \sum_{\pi \in \mathcal{P}} \exp \left( -\frac{D(\pi)}{\gamma} \right), \quad (2)$$

where $\gamma$ is a smoothing parameter, and $\mathcal{P}$ denotes the set of all possible alignment paths (i.e., warping paths) between the two sequences. Each path $\pi$ defines a time-step correspondence that allows for non-linear alignment, accommodating temporal shifts or rate differences between SHAP patterns. The cost of a given path $\pi$ is computed as:

$$D(\pi) = \sum_{(i,j) \in \pi} d(x_i, y_j), \quad (3)$$

where $d(x_i, y_j)$ is the pairwise dissimilarity between elements from sequences $X$ and $Y$, computed using Euclidean distance. An example of Soft-DTW divergence matrices computed for four seizures of Patient 1000 is shown in Fig. 5, where each cell quantifies the dissimilarity between SHAP segments over time. These matrices reveal distinct temporal dynamics and intra-seizure structure useful for downstream clustering.

The resulting pairwise distances are clustered using the $k$-medoids algorithm [19], which minimizes intra-cluster dissimilarities using actual representative segments. K-medoids, widely used in biomedical analysis for its robustness [19], was employed for stage discovery with $k = 3$, consistent with the window-size step and reflecting clinical relevance (trigger, transient, steady stages). The optimization objective is:

$$\min \sum_{i=1}^{k} \sum_{x \in C_i} d(x, m_i), \quad (4)$$

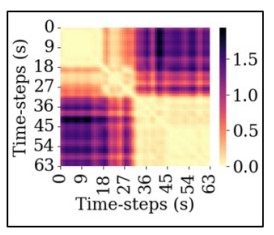
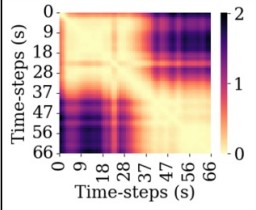

a) Seizure 1 (64x64)  b) Seizure 2 (67x67)

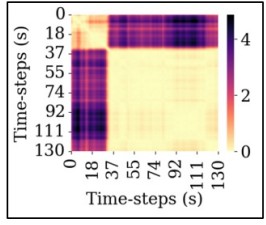
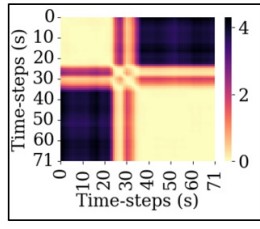

c) Seizure 3 (196x196)  d) Seizure 4 (108x108)

Fig. 5. Soft-DTW divergence matrices for Patient 1000 across four seizures. Heatmaps show pairwise divergence of SHAP segments (50% overlap); darker regions indicate higher dissimilarities.

where $C_i$ is the $i$-th cluster and $m_i \in C_i$ is the medoid (i.e., the representative segment) of cluster $C_i$. The function $d(x, m_i)$ denotes the dissimilarity between data point $x$ and medoid $m_i$.

### D. Evaluation Metrics and Visualization

We evaluate several aspects of effectiveness of our proposed methodology. For SHAP-based EZ localization (Sec. II-B), we first describe the classification performance of our patient-specific XGBoost classifiers using accuracy, F1-score, and 5-fold cross-validation, emphasizing their effectiveness in identifying seizure-relevant channels through SHAP-based interpretation. Second, for cluster-based stage discovery (Sec. II-C), we assess our clustering method using the silhouette coefficient [19], a metric quantifying cluster separability. Third, we visualize these clusters (i.e., stages) using Uniform Manifold Approximation and Projection (UMAP) [21] to illustrate distinct seizure stages clearly, and depict the temporal summaries of these stages with respect to raw SEEG waveforms. Finally, clinical alignment is evaluated by comparing model-recognized stage transitions with clinician-annotated seizure onsets. Quantitative validation of these visualizations and additional analytical results are discussed in Section III.

### III. EXPERIMENTAL RESULTS

#### A. Model Performance and EZ Localization

Using SEEG signals, we employed patient-specific XGBoost classifier to distinguish PPS and non-seizure segments, and measure performance through 5-fold cross-validation. These results confirm the classifier reliably models patient-specific brain dynamics, which supports the use of its SHAP values for downstream stage discovery. As detailed in Table I, the classifiers consistently exhibited excellent performance across all patients, demonstrating an average ROC AUC of 0.992 and an average seizure F1-score of 0.959. In some cases, such as Patient 1000 and Patient 1600 achieved near-perfect

TABLE I
XGBOOST CLASSIFIER PERFORMANCE METRICS (5-FOLD CROSS-VALIDATION AVERAGES PER PATIENT).

| Patient ID | ROC AUC | F1-score | Sensitivity | Specificity |
|---|---|---|---|---|
| 1000 | 0.997±0.002 | 0.971±0.011 | 0.967±0.019 | 0.975±0.015 |
| 1100 | 0.995±0.001 | 0.965±0.006 | 0.963±0.009 | 0.967±0.008 |
| 1300 | 0.991±0.004 | 0.954±0.017 | 0.939±0.027 | 0.972±0.006 |
| 1400 | 0.985±0.006 | 0.951±0.011 | 0.956±0.011 | 0.924±0.027 |
| 1500 | 0.985±0.006 | 0.939±0.013 | 0.920±0.029 | 0.960±0.020 |
| 1600 | 0.991±0.003 | 0.953±0.008 | 0.943±0.018 | 0.959±0.007 |
| 1700 | 0.996±0.003 | 0.969±0.014 | 0.968±0.016 | 0.970±0.020 |
| 1800 | 0.994±0.002 | 0.966±0.007 | 0.954±0.007 | 0.979±0.008 |
| **Avg.** | **0.992±0.003** | **0.959±0.011** | **0.951±0.017** | **0.963±0.014** |

ROC AUC scores (0.997 and 0.996, respectively), highlighting the robustness and accuracy of the classification models.

As tabulated in Table II, a significant agreement exists between clinician-selected channels and those identified through our data-driven SHAP approach (marked with an asterisk). Patient 1600 (with 9 seizures) exemplified this concordance, consistently highlighting clinician-identified channels RB1 and RB2 across multiple seizures. This alignment underscores the clinical validity and robustness of our automated channel-selection strategy.

#### B. Intra-Seizure Stage Discovery and Clustering

The central contribution of our approach is the recognition of intra-seizure stages (initial, transient and steady) through unsupervised clustering of temporal SHAP profiles. Table III outlines the effectiveness of our clustering approach, which applies k-medoids clustering using soft-DTW as the distance metric. Clustering performance is evaluated using silhouette scores, where SHAP-based feature clusters yielded higher average separability (0.71) compared to raw SEEG signal clusters (0.57). Notably, Patient 1000 demonstrated particularly distinct SHAP-based stages (silhouette score: 0.84) in the second seizure, where three well-separated clusters were discovered. Visualization of cluster separability using UMAP projections (Fig. 6) for this example further confirmed the superior performance of SHAP-derived features. While most cases favored SHAP-based clustering, Patient 1300 showed closely matched silhouette scores between SHAP-based and raw feature clusters (0.79 vs. 0.75; Table III), with a slight margin still supporting the SHAP-based approach. For both SHAP and raw SEEG approaches, window sizes were optimized per seizure. In general, and SHAP-derived features achieved significantly higher silhouette scores (p = 0.0039, Wilcoxon test).

Our framework also provides quantitative metrics to characterize seizure dynamics, such as the number of stage transitions and stage durations. These values, detailed in Table II, illustrate variations in seizure complexity. For example, Seizure 6 of Patient 1800 exhibited highly dynamic behavior (54 transitions), whereas seizures from Patient 1600 displayed more stable, clearly delineated stages (average 2 transitions). These metrics offer knowledge and hypotheses to clinicians to

TABLE II
INTRA-SEIZURE STAGE RECOGNITION RESULTS PER PATIENT.

| Patient ID | Seizure/Event # (Type)‡ | Duration (s) | Selected Channels | Stage Durations (s) | First Stage Offset (s)† | Stage Transitions | Best Window Size (s) |
|---|---|---|---|---|---|---|---|
| 1000 | 1 (ES) | 63 | LA1*, LA2*, LC12, LC10 | 17.0, 14.0, 32.0 | -3.0 | 2 | 2 |
| | 2 (ES) | 66 | LA1*, LA2*, LB10, LA8, LC12 | 1.0, 35.0, 30.0 | -19.0 | 3 | 2 |
| | 3 (ES) | 195 | LA1*, LA2*, LB10, LA8 | 28.5, 19.5, 147.0 | -18.5 | 3 | 3 |
| | 4 (ES) | 107 | LA1*, LA2*, LA8, LC12 | 36.0, 9.0, 61.5 | 16.0 | 4 | 3 |
| 1100 | 1 (FIC) | 295 | LB2*, LB10, LB8, LB1* | 98.0, 44.0, 153.0 | -19.0 | 21** | 2 |
| | 2 (FIC) | 199 | LB2*, LB1*, LB8, LB7 | 60.0, 59.0, 79.5 | 1.0 | 4 | 3 |
| 1300 | 1 (FPC- FIC) | 37 | LB1*, LB4, LB10, LB2* | 23.0, 14.0, 0 | -6.0 | 1 | 4 |
| | 2 (FPC- FIC) | 140 | LB1*, LC6, LC7, LC4, LC1* | 8.0, 114.0, 18.0 | -16.0 | 4 | 4 |
| | 3 (FPC- FIC) | 154 | LB1*, LC6, LC7, LC1* | 24.5, 27.0, 102.0 | 1.0 | 7 | 3 |
| 1400 | 1 (FPC) | 72 | RI7*, RA2*, RI8, RC10 | 8.0, 12.0, 52.0 | -18.0 | 9 | 2 |
| | 2 (ES) | 102 | RI7*, RA2*, RI1, RI8 | 11.0, 55.0, 36.0 | -19.0 | 21** | 2 |
| | 3 (ES) | 83 | RI7*, RA2*, RI8, RB5 | 12.0, 47.0, 24.0 | -14.0 | 21** | 2 |
| 1500 | 1 (ES) | 52 | LB9, LB7, LB1*, LB8 | 2.0, 29.0, 21.0 | -17.0 | 7 | 2 |
| | 2 (ES) | 69 | LB9, LB2*, LB3, LB5 | 6.0, 31.0, 32.0 | -17.0 | 16** | 2 |
| | 3 (FBTC) | 144 | LB9, LB2*, LB1*, LB10 | 23.0, 19.0, 101.5 | -1.0 | 2 | 2 |
| 1600 | 1 (FPC) | 64 | RB1*, RB2*, RB8, RB4 | 24.0, 14.0, 26.0 | -1.0 | 4 | 2 |
| | 2 (FPC) | 82 | RB1*, RB2*, RB6, RB9 | 18.0, 41.5, 22.5 | -2.0 | 2 | 3 |
| | 3 (FPC) | 67 | RB1*, RB2*, RB3, RB9 | 20.0, 24.0, 23.0 | 0.0 | 2 | 2 |
| | 4 (FPC) | 65 | RB1*, RB2*, RB6, RB9 | 18.0, 23.0, 24.0 | -2.0 | 2 | 3 |
| | 5 (FPC) | 69 | RB1*, RB2*, RB6, RB9 | 18.0, 27.0, 24.0 | -2.0 | 3 | 2 |
| | 6 (FPC) | 69 | RB1*, RB2*, RB4, RB6 | 20.0, 25.0, 24.0 | 0.0 | 2 | 2 |
| | 7 (FPC) | 69 | RB1*, RB10, RB2*, RB6 | 19.0, 23.0, 27.0 | -1.0 | 2 | 2 |
| | 8 (FPC) | 84 | RB1*, RB2*, RB9, RB4 | 18.0, 19.5, 46.5 | -2.0 | 2 | 3 |
| | 9 (FPC) | 75 | RB1*, RB2*, RB10, RB9 | 19.0, 25.5, 30.0 | -1.0 | 2 | 2 |
| 1700 | 1 (FPC) | 85 | LB1*, LB3, LB7, LB5 | 18.0, 43.5, 23.5 | -2.0 | 2 | 3 |
| | 2 (FIC) | 90 | LB1*, LB3, LB7, LB10 | 14.0, 5.0, 71.0 | -6.0 | 2 | 2 |
| | 3 (FIC) | 94 | LB1*, LB7, LB3, LB10 | 21.0, 6.5, 66.0 | 1.0 | 2 | 3 |
| 1800 | 1 (ES) | 47 | LC2*, LB1*, LC3*, LB7 | 12.0, 8.0, 27.0 | -15.0 | 3 | 2 |
| | 2 (FIC) | 37 | LC3*, LB3, LB10, LB1*, LC10, LC2* | 23.0, 14.0, 0 | -14.0 | 3 | 4 |
| | 3 (FIC) | 56 | LC2*, LB1*, LC3*, LB3 | 5.0, 16.0, 35.0 | -10.0 | 3 | 2 |
| | 4 (FBTC) | 76 | LC10, LB10, LB3, LB1* | 13.0, 12.0, 51.0 | -18.0 | 6 | 2 |
| | 5 (FIC) | 24 | LB10, LB3, LB8, LB1* | 15.0, 9.0, 0 | -18.5 | 2 | 3 |
| | 6 (FIC) | 170 | LB3, LB10, LB1*, LC10, LB4, LC3* | 22.0, 83.0, 65.0 | -14.0 | 54** | 2 |
| | 7 (FIC) | 22 | LB1*, LB3, LC2*, LB8 | 11.0, 6.0, 5.0 | -10.0 | 3 | 2 |
| | 8 (ES) | 369 | LB10, LC2*, LB1*, LB3, LC3* | 270.0, 93.0, 6.0 | 2.0 | 4 | 4 |
| | 9 (ES) | 38 | LB10, LB3, LB1*, LC10 | 19.0, 13.0, 6.0 | -14.0 | 4 | 2 |
| | 10 (FPC) | 45 | LB1*, LB10, LB3, LB8 | 14.0, 26.0, 5.0 | 2.0 | 6 | 4 |
| | 11 (FPC) | 70 | LB3, LB10, LB1*, LC10, LB4, LB8 | 2.0, 67.5, 0 | 0.0 | 2 | 4 |

* Clinician-selected channel based on provided patient information.
† Negative values indicate initial onset stage before clinician-marked seizure onset; positive values indicate initial stage after onset.
** Elevated transition counts may reflect seizure complexity, signal artifacts, or anatomical variability.
‡ ES = Electrographic Seizure; FPC = Focal Preserved Consciousness; FIC = Focal Impaired Consciousness; FBTC = Focal to Bilateral Tonic Clonic.

TABLE III
PATIENT-LEVEL CLUSTERING PERFORMANCE.

| Patient ID | Avg. Silhouette (**Our Work**) | Avg. Silhouette (Raw) | Avg. First Stage Offset (s) | Avg. Closest Trans. Offset (s) |
|---|---|---|---|---|
| 1000 | 0.84 | 0.40 | -6.13 | 7.50 |
| 1100 | 0.66 | 0.47 | -9.00 | 0.00 |
| 1300 | 0.79 | 0.75 | -7.00 | -0.33 |
| 1400 | 0.59 | 0.46 | -17.00 | 0.67 |
| 1500 | 0.54 | 0.47 | -11.67 | 0.00 |
| 1600 | 0.83 | 0.70 | -1.22 | -1.11 |
| 1700 | 0.78 | 0.68 | -2.33 | -0.67 |
| 1800 | 0.63 | 0.60 | -11.23 | -0.41 |
| **Avg.** | **0.71** | **0.57** | **-8.20** | **0.71** |

quantify seizure complexity and severity, potentially guiding clinical interpretations and treatment strategies.

### C. Temporal Pattern of Seizures

It is understood that identifying seizure onset can sometimes be visually challenging and ambiguous. One goal of this study is to evaluate how well the seizure onset, found by our method align with clinical seizure onset. We intentionally considered 20 seconds before the clinician's selected onset to allow our computational model explore the pattern. We analyzed the

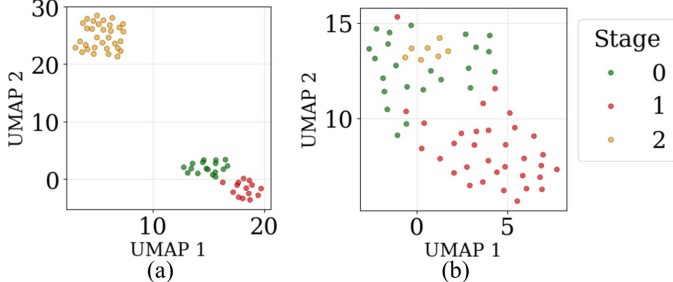

Fig. 6. UMAP of seizure stages for Patient 1000 (Stage 0: trigger, Stage 1: transient, Stage 3: steady), Seizure 1: (a) soft-DTW features (silhouette = 0.81) show clear separation; (b) raw SEEG segments (silhouette = 0.35) expectedly showing poor separability.

temporal offset between the first data-driven stage transition and the clinician-marked onset. Results showed that in some cases, the initial transition occurred prior to clinical labeling.

Quantitatively, the closest detected stage transition occurred an average of 0.71 seconds from the clinician-annotated seizure onset across all seizures (Table III), highlighting the

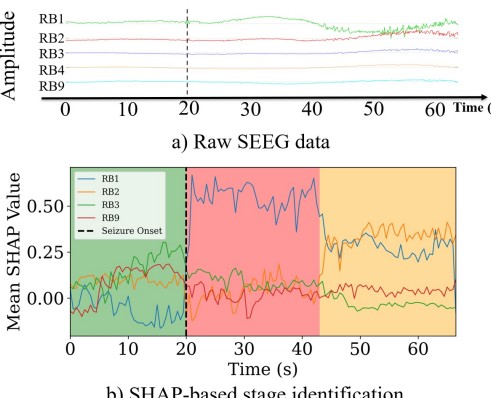

a) Raw SEEG data

b) SHAP-based stage identification

Fig. 7. Raw SEEG and stage transitions for Patient 1600, Seizure 3: (a) RB channels with onset at 20 s; (b) SHAP profiles with stage recognition (Stage 0 = green: trigger; Stage 1 = red: transient; Stage 3 = yellow: steady; dashed line = onset).

method's ability to align closely with expert visual assessment. As shown in the "First Stage Offset" column of Table III, this metric quantifies the temporal gap between the clinician-annotated onset and the first stage transition detected by the model. Note that the average offset of –8.2 seconds indicates that stage transitions that precede visually marked onset (often in range of 1–19 seconds). As visual inspection may miss early or subtle ictal activity, our method uses these annotations as temporal references, with the negative offsets reflecting early ictal dynamics that can inform anticipatory interventions and treatment timing. In addition to onset alignment, the number of stage transitions reported in Table II offers insight into the temporal structure of each seizure episode. A larger number of transitions may reflect more complex or fluctuating intra-seizure activity, whereas a smaller number indicates greater temporal stability. We may choose to filter out very short (e.g. less that 1 sec.) stages as they may be due to noise and artifact. The Stage Duration column presents the total time spent in each of the three stages (trigger, transient, and steady). This information is useful for characterizing the relative contribution of each stage and offers a more detailed analysis of seizure evolution across episodes and patients that can be utilized for personalized treatments.

Clinical interpretability was further supported by visualizing stage transitions alongside raw SEEG signals. As a representative example, Seizure 3 from Patient 1600 demonstrated a strong alignment between the data-driven stage boundaries and clinically observable changes, with the first detected stage occurring at the same times as the clinician-marked onset and a high silhouette score of 0.87. This example, shown in Fig. 7, highlights the interpretative capability of our approach when applied to raw traces. Fig. 8(a-c) illustrates an overview of the stage patterns in a few other seizures for the same patient. Across all episodes, the framework consistently detected three distinct stages with clear temporal separation, reinforcing the robustness and clinical relevance of the method. Some patients, especially those with Electrographic Seizures (ES) [22] exhibit a more variant seizure pattern. For example, three seizures for Patient 1400 shown in Fig. 8(e,f), display approximately

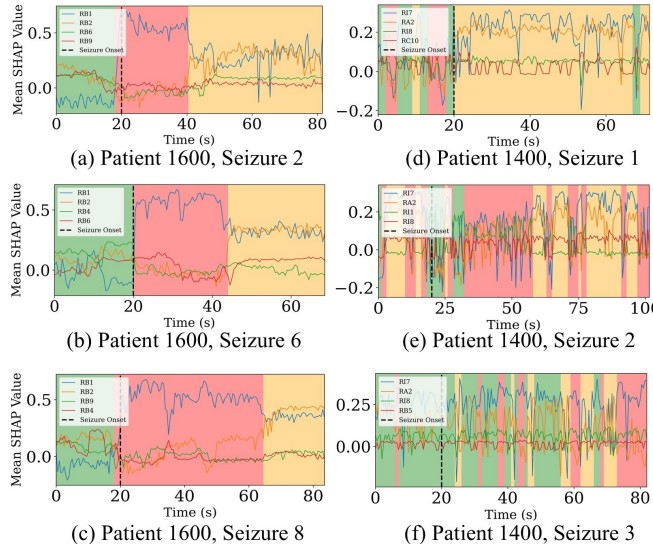

(a) Patient 1600, Seizure 2

(d) Patient 1400, Seizure 1

(b) Patient 1600, Seizure 6

(e) Patient 1400, Seizure 2

(c) Patient 1600, Seizure 8

(f) Patient 1400, Seizure 3

Fig. 8. Stage recognition across six seizures: (a–c) Patient 1600 showing consistent transitions aligned with onset; (d–f) Patient 1400 showing fragmented dynamics, with Seizures 2–3 having 21 transitions from high variability (Stage 0 = green: trigger; Stage 1 = red: transient; Stage 3 = yellow: steady).

21 stage transitions. We comment on such cases in the next Subsection.

### D. Factors Impacting Stage Recognition

This section explores critical factors affecting the accuracy and interpretability of intra-seizure stage recognition. Specifically, we discuss the impact of temporal segmentation parameters, clustering configurations, and seizure type variability.

• **Impact of window size:** To account for differences in seizure duration and complexity, we applied variable window sizes (e.g., $w = 1$-5 seconds, a typical range commonly used) when segmenting SHAP temporal profiles. As shown in Table II, the optimal window size varied not only across patients but also across seizures within the same patient. For instance, the four seizures from Patient 1000 were best modeled using different window lengths, underscoring the importance of seizure-specific temporal resolution rather than a fixed global setting.

• **Impact of cluster numbers:** We chose to fix the number of clusters at $k = 3$ for all seizures to facilitate consistent comparisons among seizures and patients. This choice aligns with commonly recognized seizure phases (onset, propagation, and symptomatic) as reported in recent SEEG-guided clinical studies [23], corresponding to our stages of trigger, transient, and steady activity.

In addition, silhouette scores for Patient 1600 are shown as an illustrative example supporting $k = 3$ as a reasonable setting (Fig. 9(b)). For Seizure 9 of the same patient, scores for $k = 2, 3$, and $4$ were comparable (Fig. 9(a)), suggesting that finer granularities may also be viable. While higher $k$ values could capture sub-stages, $k = 3$ was preferred for clinical interpretability.

• **Impact of Seizure Types:** Consistent seizure stage patterns were observed for Patient 1600, whose episodes were all classified as *Focal Preserved Consciousness* (FPC) [22].

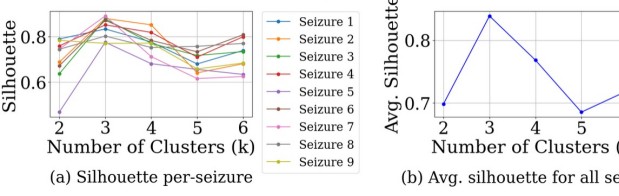

(a) Silhouette per-seizure  (b) Avg. silhouette for all seizures

Fig. 9. (a) Silhouette scores of each seizure for Patient 1600. (b) Average across nine seizures, peaking at $k = 3$.

For this patient, both spatial and temporal findings closely aligned with clinical annotations, as illustrated by Fig. 8 and Table II. Spatially, clinician-selected channels ranked within the top two of our SHAP-based analysis. Temporally, the first-detected stage transitions were aligned with the clinician-marked seizure onsets (with small offsets), and the small number of total stage transitions ($\leq 4$) showed well-identified ictal evolutions.

In contrast, Patient 1400 exhibited a more variant seizure profile. As listed in Table II, Seizure 1 was a FPC event, but Seizures 2 and 3 were identified as *Electrographic Seizure* (ES) [22]. These seizures displayed approximately 21 stage transitions each (see Table II, Fig. 8(e,f)), suggesting more complex temporal dynamics than Seizure 1 (FPC). Multiple factors could contribute to or influence this variability such as multiple generators, variable networks and anatomical locations of SEEG electrodes. Overall, this highlighted the significance of our intra-seizure analysis, as different seizure events initiated from an identical patient could show variant temporal patterns during their ictal propagation.

## IV. CONCLUSION

This work presents a patient-specific framework for intra-seizure pattern recognition using SEEG data. By integrating SHAP-based channel score, we quantify the contributions of different spatial regions. With Soft-DTW-based temporal pattern similarity and unsupervised clustering, we identify three temporal stages—onset, transient, and steady—with clinical meaningfulness. Evaluated on eight patients (38 seizures) as a pilot study, the framework shows a strong potential for personalized surgical planning and neurostimulation. Future work will validate our findings on a larger dataset and correlate stages with clinical factors to refine treatment timing and parameters.

## ACKNOWLEDGEMENT

This work was partially supported by the University of Texas at Dallas, Office of Research and Innovation, through the SPARK Grant Program.

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
