# OpenReview forum: "Intra-Seizure Pattern Recognition for Personalized Treatment"
_IEEE.org/EMBS/BHI/2025/Conference — BHI 2025_

### Official Review · Reviewer_TBps · 2025-06-26
**Intra-Seizure Pattern Recognition via SEEG: A Novel Method with Promising Results, but Gaps in the Evaluation**

**Confidence:** 4
**Clarity Of Writing:** great
**Clinical Significance:** excellent
**Methodological Novelty:** great
**Overall Rating:** 6
**Final Rating:** 7

**Experiments And Results:**

fair

**Questions For The Authors:**

These questions aim to clarify aspects of the evaluation protocol that currently limit confidence in the paper’s conclusions:

 - $\text{\textbf{On Seizure Onset Estimation}}$: How can the model’s estimation of seizure onset be formally validated against the clinician’s annotation? Is there a gold standard or medically accepted criterion that could be used to determine whether the model or the clinician is more accurate in identifying the true seizure onset? Alternatively, can the authors justify the average –8.2 second offset observed in Table III as clinically meaningful rather than erroneous?
 - $\text{\textbf{On Probe Selection and the Elbow Threshold}}$: Is it possible to provide a clinical or empirical validation of the relevance of the SEEG channels selected by the model—for example, in terms of medically backed true positive and false positive rates? In addition, can the authors demonstrate that the elbow threshold used for selecting SHAP-ranked channels was determined independently of clinician annotations, and not influenced by prior knowledge of the ground truth (e.g., to ensure that the inclusion of clinician-selected probes was not coincidental or biased)?

Convincing responses to these questions would significantly strengthen the paper and directly address its major limitations. If adequately answered—and assuming the minor issues are also addressed—this work would warrant a score in the “accept” or “strong accept” range. However, if these concerns remain unresolved, they could justify a score closer to “borderline reject.”

**Strengths:**

Here are the strengths of the paper, listed in order of importance:

- $\text{\textbf{Novel method for a clinically significant problem}}$: Unlike more popular tasks in the field, such as seizure prediction from EEG signals, intra-seizure pattern recognition is a generally overlooked area. Proposing a pipeline with tools and meaningful stages drawn from clinical practice to aid pre-surgical planning is a commendable contribution of this work.
- $\text{\textbf{Thorough explanation of the pipeline}}$: The proposed pipeline includes many steps and draws on several algorithms from the literature. Yet, the paper manages to explain each component clearly, even for readers who may not be familiar with all the methods used. The chosen figures are especially informative and significantly enhance the clarity of the explanation.
- $\text{\textbf{In-depth analysis of factors impacting stage recognition}}$: The paper studies the impact of both pipeline-dependent parameters (e.g., window size, number of clusters) and external factors (e.g., seizure type). This demonstrates the robustness of the method and strengthens the credibility of the experimental validation.

**Summary Of The Paper:**

This paper presents a patient-specific analysis pipeline designed to extract spatiotemporal insights from SEEG data collected from individuals with drug-resistant epilepsy during seizures. The goal is to support personalized pre-surgical planning by identifying intra-seizure patterns.

The proposed framework starts by training an XGBoost classifier to distinguish seizure from non-seizure periods. SHapley Additive exPlanations (SHAP) are then used to compute the contribution of each SEEG channel over time, providing a spatial-temporal importance map during a defined window. Next, SHAP profiles are segmented into time windows, and their temporal evolution is compared using Soft-DTW to quantify dissimilarities. The resulting divergence matrices serve as input to a k-medoids clustering algorithm, which partitions each seizure episode into three clinically meaningful stages.

Validated on data from 8 patients and 38 seizures, the framework demonstrates strong classification performance and meaningful intra-seizure stage discovery. The results also reveal seizure-specific variations in temporal dynamics, providing quantitative metrics such as the number of stage transitions and their timing.

**Weaknesses:**

Weaknesses of the paper have been separated between major and minor, and listed by order of importance.

Major weaknesses:
- $\text{\textbf{Discrepancy between clinician and model in seizure onset estimation:}}$: The conclusion states that the method "aligns well with clinician-annotated onsets," yet Section III-C and Table III clearly show notable differences between the model-identified and clinician-labeled seizure onsets, with an average offset of –8.2 seconds and substantial inter- and intra-patient variability (see Table II). This discrepancy raises concerns about which onset estimate is more reliable. However, the paper does not provide any ground truth evaluation or gold standard against which the model’s or the clinician’s estimates can be validated. This limits the interpretability of the claimed alignment and leaves open the possibility of systematic errors on either side.
- $\text{\textbf{Lack of evaluation of the subjective elbow method}}$: The selection of relevant SEEG channels relies on the elbow method (ref. [16] in the paper), which is known to be subjective and sensitive to minor variations in input data. Despite this, the paper does not explore the stability of the elbow threshold or its impact on downstream results. Table II also suggests that clinician-selected channels often appear near the threshold, which could imply manual tuning to reduce false negatives. Moreover, the paper does not assess whether all selected channels contribute meaningfully to model performance, leaving open the question of whether some may be false positives.
- $\text{\textbf{Insufficient positioning within the existing literature}}$: The paper claims that the literature lacks a "systematic analysis of intra-seizure patterns in SEEG data." However, recent studies [1–3] have addressed closely related problems involving dynamic seizure modeling and SEEG-based seizure analysis. While the presented methodology is original and well-constructed, a more thorough comparison with these prior works would help contextualize the contribution and better guide the reader through the landscape of current research.

Minor weaknesses:
- The paragraph "Understanding intra-seizure patterns [...] involving symptomatogenic regions." in the introduction discusses clinical practices and implications but lacks proper citations for its claims.
- Several in-text references to figures and tables appear broken (e.g., the first mention of Fig. 1 in Section II).
- Performance metrics of the XGBoost classifier are reported as average values over 5-fold cross-validation, but standard deviations are omitted. Including them would provide insight into the model’s stability.
- The placement of Figs. 6–8 and Table III interrupts the narrative flow and could be refined to improve readability.
- In several figures (Figs. 4–9), the font size for axis labels and annotations is too small, making them difficult to read

[1] Peng, Genchang, Mehrdad Nourani, and Jay Harvey. "SEEG-Based Bilateral Seizure Network Analysis for Neurostimulation Treatment." IEEE Transactions on Neural Systems and Rehabilitation Engineering (2025).

[2] Ramos, Alejandro Nieto, et al. "Epileptic network identification: insights from dynamic mode decomposition of sEEG data." Journal of Neural Engineering 21.4 (2024): 046061.

[3] Sun, Jie, et al. "Seizure pathways changes at the subject-specific level via dynamic step effective network analysis." IEEE Transactions on Neural Systems and Rehabilitation Engineering 32 (2024): 1324-1332.

---

### Official Review · Reviewer_JnqD · 2025-07-18
**Intra-Seizure Pattern Recognition for Personalized Treatment**

**Confidence:** 4
**Clarity Of Writing:** fair
**Clinical Significance:** good
**Methodological Novelty:** fair
**Overall Rating:** 6

**Experiments And Results:**

good

**Questions For The Authors:**

o	How do the three stages (trigger, transition, and steady) are quantitatively defined? For instance, what is the threshold for some intra-seizure activity to be considered steady?

**Strengths:**

o	The proposed topic of looking at intra-seizure patterns is interesting and less explored.
o	The paper provides comprehensive quantitative analysis, including sensitivity analyses of the parameters.

**Summary Of The Paper:**

The paper provides a patient-specific framework for intra-seizure pattern recognition, where these patterns are clustered into three stages: trigger, transition, and steady.

**Weaknesses:**

o	A lot of details in the paper are missing, which makes it difficult to understand the proposed method. For example, what do the SHAP values represent in this context?
o	The reasoning behind choosing different methods is not clearly explained. For example, why was the K-medoid method selected instead of other clustering methods?

---

### Official Review · Reviewer_9s8r · 2025-07-19
**Ambiguities in the Methodology**

**Confidence:** 5
**Clarity Of Writing:** great
**Clinical Significance:** fair
**Methodological Novelty:** great
**Overall Rating:** 4

**Experiments And Results:**

good

**Questions For The Authors:**

Selecting optimal window size requires existing clusters. Are you pre-clustering with k=3 for each window size, or running k-medoids clustering 5 times per seizure just to select the optimal window?

The mention of "0.5-second step size" appears only in Figure 5's caption. Is this step size used for all window sizes? If so, a 5-second window with 0.5-second steps would heavily overlap (90% overlap). Is this intentional?

**Strengths:**

Solid work on an important clinical problem. Interesting idea to use DWT for feature extraction and SHAP values as temporal biomarkers.

**Summary Of The Paper:**

The paper introduces a patient-specific computational framework for analyzing SEEG data to identify clinically meaningful seizure stages in epilepsy patients.

**Weaknesses:**

The paper mentions using DWT but doesn’t provide implementation details. Which wavelet family? How many decomposition levels? Are you using approximation coefficients, detail coefficients, or both? What's the final feature dimensionality per channel?

You report high classification performance in Table I for distinguishing seizure from non-seizure segments, using the SHAP-based approach but where's the comparison with classification using raw SEEG data? Without any baseline, the classification results only show that seizures are distinguishable from non-seizures.

In Table III, you're selecting the optimal channels and the best window size (highest silhouette) for each seizure of every patient, and comparing their clustering performance against raw data using unoptimized fixed parameters (36 channels, 2-second windows?). How is this a fair comparison to support the effectiveness of the SHAP-based approach?
To show the performance improvements in Table III, why not use paired tests rather than comparing averages without any statistical comparison?

The paper doesn’t clearly document the path to the "consistent" identification of k=3 stages across all patients.

There are no validation/references that these 3 stages are clinically meaningful beyond timing, and no concrete explanation of how identifying these stages changes treatment. The discussion completely sidesteps the core question of clinical utility.

The reference list seems outdated for a 2025 paper. Most computational methods cited are from 2017-2019, missing the most recent advances in deep learning approaches for seizure analysis.

---

### Official Review · Reviewer_gPSA · 2025-07-20
**Intra-Seizure Pattern Recognition for Personalized Treatment**

**Confidence:** 3
**Clarity Of Writing:** fair
**Clinical Significance:** good
**Methodological Novelty:** great
**Overall Rating:** 6
**Final Rating:** 7

**Experiments And Results:**

great

**Questions For The Authors:**

In Section 3A, the authors state that “we employed patient-specific XGBoost classifiers to distinguish two classes of PPS and nonseizure segments”. This raises the following questions:
Is the XGBoost trained per seizure type, or are all seizures pooled into one model per patient, and are the calculated SHAP values seizure-specific? The paper lacks a clear explanation of how seizure type is incorporated into the XGBoost training.

**Strengths:**

1. Methods: The authors integrate SHAP value interpretation with time-series alignment techniques (Soft-DTW) and unsupervised learning (clustering) to identify temporal features in SEEG data, which is a novel method.

2. Visualization: The paper provides clear visualizations to help identify SHAP-based stage identification.

3. Clinical Relevance: The authors compare their results and clinical onset of stage transitions, which shows clinical relevance.

4. Discussion: The discussion includes an analysis of the effects of different window sizes and the number of clusters to support reproducibility.

**Summary Of The Paper:**

This paper proposed a patient-specific framework to analyze intra-seizure patterns of SEEG data, in order to improve personalized treatment for patients. The author used XGBoost classifier to obtain SHAP values to quantify the contribution of each SEEG channels and applied the DTW to segment temporal SHAP trajectories. Then the authors performed clustering to identify three intra-seizure stages: trigger, transient, and steady. This proposed framework is validated on eight patients and demonstrates high classification performance and seizure stage localization.

**Weaknesses:**

1. Clarity of Data Source: This paper lacks a clear description of the data used. It is unclear whether the SEEG recordings of eight patients were self-collected or from a public dataset. And no information regarding data collection protocols and patient demographics is provided, which limits the reproducibility of the study.

2. Clarity of the Method: The description of XGBoost classification is vague in the following aspects:
- What data is used for training the XGBoost ( e.g., is it the SEEG recordings from one session, or from multiple sessions, or the T segments?)
- Is the train-test data splitting controlled for each seizure type or across seizures?

3. Comparison to previous work: The authors point out the flaw in prior works that “treat different seizure type equivalently”; however, it is not clear about the benefits of modeling seizure types separately in the paper. The authors compare the clustering results (stage identification) between soft-DTW features and raw SEEG signal - not between modeling seizures separately versus treating them equivalently, thus lacking a demonstration of the benefit of intra-seizure pattern.

4. Limited sample size: The study only includes eight patients, which may be sufficient for a pilot study but limits the generalizability.